# A Fast Weighted Fuzzy C-Medoids Clustering for Time Series Data Based on P-Splines

**DOI:** 10.3390/s22166163

**Published:** 2022-08-17

**Authors:** Jiucheng Xu, Qinchen Hou, Kanglin Qu, Yuanhao Sun, Xiangru Meng

**Affiliations:** 1College of Computer and Information Engineering, Henan Normal University, Xinxiang 453007, China; 2Engineering Lab of Intelligence Business & Internet of Things, Xinxiang 453007, China

**Keywords:** fuzzy C-medoids, weight fuzzy clustering analysis, similarity measure, time series, P-splines

## Abstract

The rapid growth of digital information has produced massive amounts of time series data on rich features and most time series data are noisy and contain some outlier samples, which leads to a decline in the clustering effect. To efficiently discover the hidden statistical information about the data, a fast weighted fuzzy C-medoids clustering algorithm based on P-splines (PS-WFCMdd) is proposed for time series datasets in this study. Specifically, the P-spline method is used to fit the functional data related to the original time series data, and the obtained smooth-fitting data is used as the input of the clustering algorithm to enhance the ability to process the data set during the clustering process. Then, we define a new weighted method to further avoid the influence of outlier sample points in the weighted fuzzy C-medoids clustering process, to improve the robustness of our algorithm. We propose using the third version of mueen’s algorithm for similarity search (MASS 3) to measure the similarity between time series quickly and accurately, to further improve the clustering efficiency. Our new algorithm is compared with several other time series clustering algorithms, and the performance of the algorithm is evaluated experimentally on different types of time series examples. The experimental results show that our new method can speed up data processing and the comprehensive performance of each clustering evaluation index are relatively good.

## 1. Introduction

With the rapid development of computer information technology, the research on time series clustering is gradually rising in various fields such as finance, biology, medicine, meteorology, electricity, industry, and agriculture [1,2,3]. To better summarize and discover the effective information on these time series data, time series clustering technology has received extensive attention from researchers [4]. Through clustering, we can mine the key hidden information in time series data and find some regularities between time series sample points. It provides indispensable preparation for forecasting and anomaly detection of time series data. Consequently, the study of time series clustering is of great significance.

The data is often imperfect. Due to some human errors, machine failures, or unavoidable natural influences, noise points and outlier samples may appear in the data, this phenomenon also exists in time series data.To better avoid the impact of abnormal fluctuations in time series data onto clustering, some researchers try to smooth the data before clustering. Abraha et al. (2003) [5] proposed using B-spline fitting function data and using a K-means algorithm to divide the estimated model coefficients, which proved the strong consistency and effectiveness of the clustering method. Iorio et al. (2016) [6] used a P-spline smoother to model the raw data and then used the k-means algorithm to cluster the fitted data objects according to the best spline coefficients. D’Urso et al. (2021) [7,8] applied the B-spline fitting coefficient to the robust fuzzy clustering method and proposed four different robust fuzzy clustering methods of time series data, then applied them to the clustering of COVID-19 time series. The above studies all smooth the data to reduce the negative effects of noise. In this paper, we fit the time series in the original data using P-splines [9]. P-spline shrinks the difference penalty based on B-spline. This method will be described in detail in Section 2.1.

With the extensive research on time series clustering, many targeted methods have been proposed, such as model-based clustering, feature-based clustering, deep learning-based clustering, and fuzzy theory-based clustering. Among them, the Fuzzy C-Means (FCM) algorithm is the classical representative of fuzzy clustering [10], which adds the fuzzy concept to the k-means algorithm, and is widely used in the clustering of time series [11]. The membership value between the sample point and the cluster center is calculated to indicate the closeness of each sample point in the cluster center [12]. The FCM algorithm provides a more flexible clustering criterion so that time series samples with greater similarity can also be classified into the class to which they belong [13,14]. Compared with the K-means algorithm, FCM provides a better clustering effect [15,16]. In addition, different fuzzy systems are also applied to time series data. Pattanayak et al. (2020) [17] used FCM to determine intervals of unequal lengths, and the support vector machine also considered the membership value when modeling fuzzy logical relationships. Kumar et al. (2021) [18] proposed the nested Particle Swarm Optimization algorithm and exhaustive search Particle Swarm Optimization algorithm for fuzzy time series prediction. Xian et al. (2021) [19] proposed a fuzzy c-means clustering method based on n-Pythagorean fuzzy sets, and effectively improved the prediction accuracy by using the clustering results. Al-qaness et al. (2022) [20,21] proposed optimizing the adaptive neuro-fuzzy inference system using the enhanced marine predator algorithm, and then again proposed combining the improved Aquila optimizer with adversarial-based learning techniques to optimize the adaptive neuro-fuzzy inference system. These fuzzy systems have excellent performance on time series data, which shows that the research on time series based on fuzzy theory has far-reaching significance.

Due to the high-dimension and complexity of time series data, there will inevitably be some noise sample points in the dataset, which affects the updating of fuzzy clustering centers. To solve this problem, a new clustering algorithm is derived based on FCM, called Fuzzy C-medoids (FCMdd) [22,23]. The center of each cluster in the FCMdd algorithm is a real object in the selected data set, which greatly reduces the influence of noise data onto the new medoids, and can better deal with the problem that the FCM algorithm is sensitive to noise data [24,25,26]. Therefore, FCMdd can obtain higher quality clustering results, which makes many researchers apply FCMdd to time series clustering [27]. Izakian H et al. (2015) [28] proposed a hybrid technique using the advantages of FCM and FCMdd and used dynamic time warping distance to cluster time series. Liu et al. (2018) [29] proposed two incremental FCMdd algorithms based on dynamic time warping distance, which can easily obtain higher-quality clustering results in the face of time-shifted time series data. D’Urso P et al. (2021) [30] clustered multivariate financial time series through an appropriate cutting process, in which the Medoids surrounding partition strategy combined with dynamic time warping distances was used in the clustering process. The above studies have improved the performance of the FCMdd algorithm in different aspects, but they did not consider the impact of outlier samples on the FCMdd algorithm. In this paper, We define a new method to update the weights of each sample to improve the FCMdd algorithm, which reduces the impact of outlier samples on the clustering process and improves the robustness of our algorithm.

A robust time series clustering method not only handles the adverse effects of noise in the dataset, but also classifies sample points quickly. One of the main influencing factors of time series clustering is the complexity of the similarity metric for time series [31]. So far, many effective similarity methods have been studied for time series data [32], among which two main types are used. One is lock-step measures, its main representative method is Euclidean Distance (ED), the other is elastic measures, its main representative methods are Dynamic Time Warping (DTW) [33] and Time Warp Edit Distance (TWED) [34]. ED is simple and common, but it is susceptible to the time offset and has a small range of applications; DTW is a well-known time series similarity measurement criterion [35], and it is also the most widely used and improved algorithm so far, but its computational cost is $O\left(N^{2}\right)$ when the sample object length is *N*. Among the edit distances, there is the TWED algorithm that satisfies the triangular inequality relationship, which can deal with time series of different sampling rates, and is also an excellent time series similarity measurement method, but the time complexity is the same as DTW. Among these similarity measures, DTW and TWED are popular, but their high time complexity leads to low clustering efficiency. This paper chooses the MASS 3 algorithm [36] in the matrix configuration file to measure the similarity between time series, which has certain robustness and can improve the accuracy of time series clustering. Moreover, the matrix configuration file has some outstanding advantages, such as simplicity, no parameters, time and space complexity are relatively low [37].

In response to the problems discussed above, we propose a fast weighted fuzzy C-medoids clustering for time series data based on P-splines, named the PS-WFCMdd. PS-WFCMdd can well deal with the negative problems caused by noise and outliers in time series data, and has fast and efficient clustering performance. The main contributions of this paper include the following three aspects:Abnormal fluctuations and noises in time series data usually make it challenging to define a suitable clustering model and increase the burden of related measurements. To solve these problems, we use the P-spline to fit clustering data. Smoothing the fitted data reduces abnormal fluctuations in the data compared to the original data. At the same time, the influence of noise on fuzzy clustering is avoided. Taking the fitting data as the input of the clustering algorithm, it is easier to find similar sample points in the clustering process;In the traditional fuzzy clustering method, no matter how far the sample point is from the center of the data cluster, the membership degree of the sample point belonging to this class is not 0. This makes FCM vulnerable to boundary samples. In the proposed PS-WFCMdd, we design a new weighted fuzzy C-medoids, which can reduce the influence of outlier sample points on cluster center update by defining a new weighting method;Among the time series similarity metrics, ED is simple and fast, but it is vulnerable to time offset. DTW algorithm has high time complexity and low efficiency in the face of a large-scale time-series data model. As a time-series measurement standard, MASS 3 has the characteristics of fast computing speed and small space overhead due to its unique segmented computing method. Therefore, we propose to apply MASS 3 to weighted fuzzy C-medoids to measure the similarity between time series data and medoids more accurately and quickly.

The rest of the paper is organized as follows: in Section 2, we introduce some basic knowledge used in our algorithm. Section 3 presents our method. The experimental results are presented in Section 4, which also includes a discussion. Finally, the conclusions are presented in Section 5.

## 2. Background and Related Work

### 2.1. Time Series: A P-Spline Representation

B-splines are formed by connecting polynomial segments in a specific way. As an excellent curve representation method, it can maintain the continuity and local shape controllability of the fitted data after fitting the time series [38].

Suppose the set containing *N* time series data is $T=\left\{T_{1},T_{2},\cdots T_{i},\cdots T_{N}\right\}$, where $T_{i}=\left\{T_{i}^{1},T_{i}^{2},\cdots ,T_{i}^{j}\right\}$, let $B_{h}\left(T_{i}^{j};p\right)$ denotes the value at $T_{i}^{j}$ of the *h*-th B-spline of degree *p* for a given equidistant grid of knots. Based on the above conditions, the vector set $L\left(T_{i}\right)$ of the *j* fitting coefficients corresponding to the *i*-th time series can be obtained:(1)$$L\left(T_{i}\right)=\sum_{h}b_{h}B_{h}\left(T_{i};p\right)+\theta ,i=1,\ldots ,N,$$
where θ is the error vector and $b_{h}$ is the B-spline coefficient estimated through simple linear least-squares.

B-splines perform well for nonparametric modeling problems but have limited control over smoothness and data fitting. In this regard, Eilers and Marx [9] use a relatively large number of nodes and different penalties for the coefficients of adjacent B-splines, showing the connection of the common spline penalty on the integral of the squared second derivative. The difference penalty is integrated into the B-spline basis, which created P-splines. If a dense set of splines is used, the curve is obtained by minimizing ${∥L\left(T_{i}\right)-Bb∥}^{2}$. An element of B is $B_{h}\left(T_{i}^{j};p\right)$, compared with the original sequence data, the fitting curve results obtained by b are quite different. To avoid overfitting, b can be estimated in a penalized regression setting:(2)$$\hat{b}=\underset{b}{\mathrm{argmin}}{∥L-Bb∥}^{2}+\lambda {∥D_{d}b∥}^{2},$$
where $D_{d}$ is a *d*-th order difference penalty matrix such that $\Delta^{d}$. The parameter λ continuously controls the smoothness of the final fit. Overfitting occurs when the smoothing parameter approaches zero, while for high values of λ, the result of the fitted function will be close to the polynomial model. The optimal spline coefficients follow as:(3)$$\hat{b}={\left(B^{⊤}B+\lambda D^{⊤}D\right)}^{-1}B^{⊤}y$$

Based on the above, the linear least-squares fitting model based on P-spline can be obtained by using differential penalty for the coefficients of the B-spline curve, which is as follows:(4)$$L\left(T_{i}\right)=\sum_{h}{\hat{b}}_{h}B_{h}\left(T_{i};p\right)+\theta ,i=1,\ldots ,N.$$

In a broad sense, the P-spline curve is limited by polynomial curve fitting and extended on the basis of a linear regression model. P-splines have many useful properties: P-splines show no boundary effects, just like many types of kernel smoothers; P-splines can fit polynomial data exactly; P-splines can conserve moments of the data.

We show a comparison of the two time series results with different fluctuations after P-spline fitting in Figure 1. Here, we can see that whether it is noisy time series A or time series B with slight irregularity, the fitting curve obtained by fitting the time series with P-spline is smoother than the original curve, but it reasonably retains the key characteristics of the time series. Therefore, this paper proposes to use P-spline to fit the time series first, which can eliminate the noise peak contained in the time series, but retain the contour and trend of the sample. Then, the fitting data is used as the input of clustering to improve the final clustering effect.

### 2.2. Weighted FCMdd

The FCMdd algorithm can objectively assign each data object to a certain cluster with a certain value of membership, thereby generating fuzzy clusters. Compared with the clustering method of the k-medoids algorithm that directly divides the sample categories, the FCMdd algorithm can reasonably perform uncertainty analysis on the categories of time series. In fact, the objective function of FCMdd is the same as that of FCM, except that *V* in the clustering process is selected from the real sample space, rather than the points in the *C* continuous spaces generated by the algorithm as the new clustering center.

Suppose the set containing *N* time series data is $T=\left\{T_{1},T_{2},\cdots T_{i},\cdots ,T_{N}\right\}$, $V=\left\{v_{1},v_{2},\cdots ,v_{C}\right\}$, $v_{i}\in T$ represents the cluster centers obtained during the clustering process. Then, the objective function of FCMdd is defined as:(5)$$\left\{\begin{array}{c} J_{m}\left(U,V\right)=\sum_{k=1}^{C}\sum_{i=1}^{N}u_{ik}^{m}d_{ik}\left(T_{i},v_{k}\right)\\ s.t.\;0\leq u_{ij}\leq 1,\sum_{k=1}^{C}u_{ij}=1 \end{array}\right.$$
where *C* represents the number of clusters, respectively; $u_{ik}$ represents the membership degree of the sample to the *k*-th clustering; *m* is a fuzzification parameter used to control the degree of clustering fuzziness. The larger the value of *m*, the wider the fuzzy decision boundary, its default value is 2; *U* is the numbership matrix; *V* is the set of cluster centers. The definition of $d_{ik}\left(T_{i},v_{k}\right)$ represents the similarity between the object $T_{i}$ and the cluster center point $v_{k}$. Both the cluster assignment results in k-medoids and the update results of medoids are discrete, while the cluster assignment results in FCMdd are continuous variables. For a randomly initialized medoid group or an updated medoid group, FCMdd can update $u_{ik}$ for the existing *C* medoid members. The update method is as follows:(6)$$u_{ik}={\left[\sum_{l=1}^{C}{\left(\frac{d_{ik}\left(T_{i},v_{k}\right)}{d_{lk}\left(T_{i},v_{l}\right)}\right)}^{\frac{1}{m-1}}\right]}^{-1}.$$

Any clustering algorithm needs to select the best cluster center. For FCMdd, selecting the best instance object as the medoid is one of the cores of the algorithm. The commonest approach is to select the object with the smallest distance from all objects in the dataset as the new medoid, but this approach will make the time complexity of the algorithm very high. We consider that only *q* sample points that maximize the membership in each cluster should be considered as new medoids. In this way, the update formula of cluster center $v_{k}$ can be defined as:(7)$$v_{c}=\min_{x\in \xi }\sum_{j=1}^{N}u_{cj}^{m}d_{cj}\left(T_{j},T\right)$$
where ξ is the set of *q* medoid candidates.

The weighted Fuzzy C-Medoids clustering algorithm (WFCMdd) obtains $w_{i}$ by weighting each data object $T_{i}$ and adding them to the objective function. The objective function of the minimization of WFCMdd can be defined as follows:(8)$$\left\{\begin{array}{c} J_{m}\left(U,V\right)=\sum_{k=1}^{C}\sum_{i=1}^{N}u_{ik}^{m}w_{i}d_{ik}\left(T_{i},v_{k}\right)\\ s.t.\;0\leq u_{ij}\leq 1,\sum_{k=1}^{C}u_{ij}=1 \end{array}\right..$$

Under constraint $\sum_{i=1}^{c}u_{ik}=1$, the value of $u_{ik}$ are the same as Equation (Equation 6). The new medoid $v_{k}$ of cluster *C* are defined as follows:(9)$$v_{k}=\min_{x\in \xi }\sum_{j=1}^{N}u_{jk}^{m}w_{j}d_{jk}\left(T_{j},T\right).$$

### 2.3. MASS 3

MASS is an algorithm for creating distance distributions for queries on time series. Using it to compute data requires only a small space overhead, with a small constant factor, allowing large datasets to be processed in main memory. In the continuous development of MASS, MASS 2 and MASS 3 came into being. Among them, MASS 3 is the best, it uses fast Fourier transform (FFT) to achieve semi-convolution, which is used to calculate all sliding dot products between query and sliding window. In order to avoid the problem of numerical error in convolution calculation when the data sample length n is very large, MASS 3 processes the data samples in segments instead of all n sample points at one time. The segments must overlap by exactly *m*-1 samples to maintain the continuity of the distance profile. Where the size of the block is a power of 2, and the size of the last block may not be a power of 2. MASS 3 does not depend on the query length (*m*), nor does it have the problem of dimension disaster, and its calculation cost is $O\left(N\log(N)\right)$.

The specific calculation steps are as follows:Scan the time series with a sliding window.Z-Normalize the window to get *Q*.The standard deviations of moving windows of a fixed size can be calculated in one linear scan. We assume *T* is the time series. Calculate cumulative sums of *T* and $T^{2}$ and store. Subtract the two cumulative sums to obtain the sum over any window, and finally calculate the standard deviation for all windows. We can obtain the expression for the standard deviation as:
(10)$$\sigma_{i}=\sqrt{\frac{C_{i+m}^{2}-C_{i}^{2}}{m}-{\left(\frac{C_{i+m}-C_{i}}{m}\right)}^{2}}$$
where: $C=\sum T,\;\;C^{2}=\sum T^{2}$.Use a half convolution to compute all sliding dot products between the query and the sliding window. If *x* and *y* are vectors of polynomial coefficients, the expression for this half convolution is:
(11)$$\mathrm{Half}\;\;\mathrm{conf}\left(x,y\right)=ifft\left(fft\left(x\right)fft\left(y\right)\right).$$The final vectorized working formula is:
(12)$$M\left(\hat{x},\hat{y}\right)=\sqrt{2\left(m-\frac{\mathrm{Half}\;\;\mathrm{conf}\left(T,Q\right)}{\sigma_{T}}\right)}.$$

## 3. Weighted Fuzzy C-Medoids Clustering Based on P-Splines

In the process of FCMdd algorithm clustering, after classifying an outlier sample into a certain cluster, in the next iterative update, because of the constraint condition A, the outlier sample will obtain a higher value of membership. In turn, the selected medoids are biased, and finally affect the clustering results. Although the traditional WFCMdd assigns different weights to each data sample point in the clustering process to obtain a better clustering effect. However, this method does not solve the negative impact of outlier samples on updating medoids. To solve this problem, we propose to judge the importance between different data and new medoids according to the distance. This increases the importance of data in more dense areas in the data space and reduces the importance of peripheral data (outlier data). To solve this problem, after the initial medoids *V* is obtained, the update weight $W_{i}$ of the sample point $x_{i}$ and each cluster center $v_{k}$ are obtained as:(13)$$W_{i}=\sum_{k=1}^{C}\exp\left(-\frac{∥x_{k}-v_{i}∥}{std\left(x_{i},v_{k}\right)}\right)$$
where std is the standard deviation between the calculated input data. By updating the weights in this way, so that the data points are closer to the $v_{k}$ point, then $W_{i}$ will be larger, and $W_{i}$ will be smaller on the contrary.

Based on the above research contents, we propose a fast weighted fuzzy C-medoids algorithm. The specific process of PS-WFCMdd is shown in Figure 2. The input to the algorithm is the original time series dataset $T=\left\{T_{1},T_{2},\cdots T_{i},\cdots T_{N}\right\}$. Then, use P-spline to fit each sample point in *T* to obtain the fitted data set $L=\left(L\left(T_{1}\right),L\left(T_{2}\right),\ldots L\left(T_{i}\right),\ldots L\left(T_{N}\right)\right)$. The P-spline will smooth the sample while preserving the main characteristics of the sample. The obtained fitting data set *L* is used as the processing object of clustering, and initial medoids are randomly selected from it. MASS 3 is used to efficiently calculate the similarity between each fitting data object and medoids, and convert it into the corresponding fuzzy membership matrix. The C∗N fuzzy membership matrix is generated after each iteration, and the intensity of the square color in the matrix represents the membership of each sample point relative to the cluster center. In the future medoids update, to solve the problem that FCMdd is sensitive to outlier samples, we propose a new weighting method to update the medoids. To redefine the weight relationship between different data objects and new medoids according to the distance, increase the weight of data objects close to medoids, and reduce the weight of outlier data objects farther from medoids. Thus, the clustering efficiency and robustness of PS-WFCMdd are improved.

In simple terms, after transforming the time series into projection coefficients on the basis of the P-spline function, it is clustered with new weights based on WFCMdd, and MASS 3 is used to measure the similarity between time series data and medoids in the clustering process. Then, the objective function of PS-WFCMdd is defined as:(14)$$\left\{\begin{array}{c} J_{m}\left(U,V\right)=\sum_{k=1}^{C}\sum_{i=1}^{N}u_{ik}^{m}W_{i}M_{ik}\left(T_{i},v_{k}\right)\\ s.t.\;0\leq u_{ij}\leq 1,\sum_{k=1}^{C}u_{ij}=1 \end{array}\right.$$

The iterative expression to obtain the final membership matrix and medoids is:(15)$$u_{ik}={\left[\sum_{l=1}^{C}{\left(\frac{M_{ik}\left(L\left(T_{i}\right),v_{k}\right)}{M_{lk}\left(L\left(T_{i}\right),v_{l}\right)}\right)}^{\frac{1}{m+1}}\right]}^{-1},$$
(16)$$v_{k}=\min_{x\in \xi }\sum_{j=1}^{N}u_{jk}^{m}W_{j}M_{jk}\left(L\left(T_{j}\right),T\right).$$

According to Equations (Equation 4), (Equation 13), (Equation 15) and (Equation 16), the steps of PS-WFCMdd can be summarized as Algorithm 1.
**Algorithm 1** Weighted fuzzy C-medoids clustering algorithm based on P-splines(PS-WFCMdd).**Input:** 
Time series data set *T*, determine the values of parameters *C*, λ and max.iter;**Output:** 
Clustered sample class label *Y*.  1:Fit each sample in the cluster dataset *T* using Equation (Equation 4), the new fitted data is obtained as $L=\left(L\left(T_{1}\right),L\left(T_{2}\right),\ldots L\left(T_{i}\right),\ldots L\left(T_{N}\right)\right)$;  2:Set iter = 0;  3:Randomly set *C* initial medoids: $L\left(T_{i1}\right),L\left(T_{i2}\right),\ldots L\left(T_{iC}\right)$.  4:**Repeat**  5:    Store the current centroids ${\left[L\left(T_{i1}\right),L\left(T_{i2}\right),\ldots L\left(T_{iC}\right)\right]}_{\mathrm{old}\;}=L\left(T_{i1}\right),L\left(T_{i2}\right),\ldots L\left(T_{iC}\right)$;  6:    Compute the fuzzy membership matrix *U* of $T_{i}$ in clusters (*C*) using Equation (Equation 15);  7:    Update the the new centroids $L\left(T_{i1}\right),L\left(T_{i2}\right),\ldots L\left(T_{iC}\right)$ using Equation (Equation 16);  8:    Update the weight $W_{k}$ using Equation (Equation 13);  9:    iter=iter + 1;10:**Until**${\left[L\left(T_{i1}\right),L\left(T_{i2}\right),\ldots L\left(T_{iC}\right)\right]}_{\mathrm{old}\;}=L\left(T_{i1}\right),L\left(T_{i2}\right),\ldots L\left(T_{iC}\right)$ or iter=max.iter;11:Calculate the classification label (*Y*) to which the sample belongs according to the final fuzzy membership *U*.

## 4. Experimental Results

### 4.1. Experiment Preparation and Evaluation Index

To demonstrate the effectiveness of our proposed method on time series datasets of different dimensions and data lengths, in the experiments, we selected 18 real-world datasets from the open time series database UCR for evaluation. Among them, 10 common time series data, whose detailed description is shown in Table 1, and the detailed description of 8 large-scale time series data in Table 2. The length of the time series in all datasets is between 96 and 2844, the size of the dataset is between 40 and 8236, and the number of classes in the dataset is between 2 and 8.

To make the experiment more complete, the data objects we choose are diversified, and a brief introduction is given below. Car recorded the contours of four different types of cars extracted from the traffic video. Lightning2 is a transient electromagnetic event recorded by a suite of optical and radio frequency instruments. Wafer is a database of wafers recorded during the processing of silicon wafers for semiconductor manufacturing. StarLightCurves are recorded celestial brightness data over time. Car, Lightning2, Wafer, and StarLightCurves are all time-series databases constructed from collections of process control measurements recorded from various sensors. Beef, Meat, Strawberry, and Rock are the classification records of items made by various spectrometers. Beefly, Facefour, Fish, and Herring record image databases for contour problems. Both the ECG200 and ECG5000 track the electrical activity recorded during a heartbeat. ChlorineConcentration and Mallat are simulated datasets constructed by software applications. Gunpoint recorded the movement trajectory data of two actors with their hands. Hourstwenty is the dataset recorded using smart home devices.

A statement is required here, PS-WFCMdd, PS-k-means, and PS-WFCMdd-sD set the corresponding parameters of P-splines for each experimental dataset are consistent. Each series has been modeled by P-splines taking cubic bases and third-order penalties. For the P-spline modeling, a fixed smoothing parameter λ is chosen for each dataset, and λ ranges from 0.3 to 0.6. The distance metric criterion for both FCM and PS-k-means is ED. Note that the number of centers for each dataset is also fixed, here we specify the number of classes in Table 1 and Table 2 as the *C* value.

In this experiment, our proposed PS-WFCMdd algorithm will be compared with six other time series clustering algorithms: traditional FCM algorithm; K-means based on P-splines [6] (PS-k-means); based on soft-DTW [39] (it is an improvement on the DTW algorithm as a differentiable loss function which can calculate all alignment costs Soft minimum) K-means algorithm (K-means-sD algorithm) [40]; soft-DTW-based K-medoids algorithm (K-medoids-sD); K-shape algorithm based on cross-correlation distance metric [41]; to better demonstrate the superiority of MASS 3, replace MASS 3 in PS-WFCMdd with softDTW, denoted as PS-WFCMdd-sD.

In the validation step, we use three widely used criteria to measure the performance of the clustering results. Due to the insufficient penalty of the Rand coefficient, the scores of the clustering algorithms are generally relatively high, and the performance of the algorithm cannot be significantly compared. Therefore, we use the adjusted Rand coefficient (ARI) [42], which is a measure of the Rand coefficient. In an improved version, the purpose is to remove the influence of random labels on the Rand coefficient evaluation results. The Fowlkes–Mallows index (FMI) [43] can be regarded as the geometric mean between precision and recall. This index shows similarity even if the number of clusters of the noisy dataset is different from that of the source data, making FMI one of the key metrics for evaluating the effectiveness of clustering. The amount of mutual information can be understood as the amount by which the uncertainty of the other decreases when guiding one. Normalized Mutual Information (NMI) is one of the most popular information-theoretic metrics in community detection methods. NMI is a normalization of the mutual information score used to scale the results between 0 (no mutual information) and 1 (perfect correlation). These standards are briefly described below.

The ARI score range is [−1, 1], negative scores indicate poor results, and a score equal to 1 indicates that the clustering results are completely consistent with the true labels. Where TP represents the number of true positive samples (the clustering results can group similar samples into the same cluster), TN represents the number of true negative samples (clustering results can group dissimilar samples into different clusters), FP represents the number of false positive samples (that is, the clustering results classify dissimilar samples into the same cluster), FN represents the number of false negative samples (clustering results group similar samples into different clusters). The definition of the ARI is:(17)$$\mathrm{ARI}=\frac{2\cdot (TP\cdot TN-FN\cdot FP)}{\left(TP+FN)(FN+TN)+(TP+FP)(FP+TN\right)}.$$

The parameter in the expression of FMI has the same meaning as the parameter in Equation (Equation 17) and can be defined as:(18)$$\mathrm{FMI}=\frac{TP}{\sqrt{\left((TP+FP)\cdot (TP+FN)\right)}}.$$

The minimum score of the Fowlkes–Mallows index is 0, indicating that the clustering results are different from the actual class labels; the highest score is 1, indicating that all data have been correctly classified.

NMI value range is between [0, 1], and the optimal value is 1. Given class labels η, and a result cluster labels *Y* obtained by a clustering algorithm. The NMI is defined as:(19)$$\mathrm{NMI}\left(\eta ,Y\right)=2\frac{\mathrm{MI}\left(\eta ;Y\right)}{H\left(\eta \right)+H\left(Y\right)}$$
where $H\left(\cdot \right)$ means entropy; MI is given by
(20)$$\mathrm{MI}\left(\eta ,Y\right)=\sum_{i=1}^{\left|\eta \right|}\sum_{j=1}^{\left|Y\right|}\frac{\left|\eta_{i}\cap Y_{j}\right|}{N}\log\frac{N\left|\eta_{i}\cap Y_{j}\right|}{\left|\eta_{i}\right|\left|Y_{j}\right|}$$
where $\left|\eta_{i}\right|$ is the number of the samples in cluster $\eta_{i}$.

### 4.2. Experimental Comparison and Analysis

#### 4.2.1. Results of PS-WFCMdd Method on Ten Common Time Series Datasets

To confirm that the PS-WFCMdd method is effective on common low-dimensional datasets, in the first part of this subsection, we compare the performance of PS-WFCMdd and six other clustering methods on the 10 low-dimensional datasets in Table 1. Table 3, Table 4 and Table 5, they show the ARI index scores, FMI index scores, and NMI index scores of the seven clustering methods on each dataset, where the bold content represents the best value in the comparison results.

Table 3 shows the clustering results under ARI. We can see that on the Beef, GunPoint, and Meat datasets, the PS-WFCMdd algorithm has an obvious score advantage. On the BeetleFly, Car, ECG200, FaceFour, and Lightning2 datasets, the PS-WFCMdd algorithm also has the best ARI scores, but the performance is not significant. On the Fish and Herring datasets, the ARI score of the K-means-sD algorithm is the best, followed by the SRFCM algorithm, PS-WFCMdd performs the next best. There is no significant difference in the performance of PS-WFCMdd-sD and PS-WFCMdd under ARI on the four datasets FaceFour, Fish, Lightning2, and Meat datasets.

Table 4 shows the clustering results under FMI. We can see that on the Beef and Meat datasets, the PS-WFCMdd algorithm has an obvious score advantage. On the BeetleFly, Fish, and GunPoint datasets, the FMI score of the PS-WFCMdd algorithm is also the best. On the Car, FaceFour, and Herring datasets, the FMI score of the PS-WFCMdd-sD algorithm is the best, and the SRFCM algorithm has a certain advantage over the remaining algorithms. On the ECG200 dataset, the FMI score of the K-means-sD algorithm is the best. On the Lightning2 dataset, the K-shape algorithm has the best score.

Table 5 shows the clustering results under NMI. We can see that on the Beef, BeetleFly, GunPoint, Car, ECG200, FaceFour, and Meat datasets, the PS-WFCMdd algorithm has obvious score advantages. On the Fish dataset, the NMI score of the K-means-sD algorithm is also the best value. On the Car, FaceFour, and Herring datasets, the NMI score of the K-means-sD algorithm is the best, and the PS-WFCMdd algorithm has a certain advantage over the remaining algorithms.

By analyzing the results of the above three tables, it can be concluded that our algorithm can obtain relatively excellent performance under the three evaluation indicators when faced with ordinary low-dimensional time series data sets. The improved algorithm based on the K-means algorithm also achieved good results in some indicators, indicating that the performance of the algorithm should not be underestimated. Although the PS-WFCMdd-sD algorithm achieves good performance under the FMI evaluation index, the overall performance is still inferior to the PS-WFCMdd algorithm. This shows that the MASS 3 we used has good performance on low-dimensional data in terms of clustering effect and the time complexity of the PS-WFCMdd-sD algorithm is high, we will further analyze it in Section 4.4.

#### 4.2.2. Results of PS-WFCMdd Method on Nine Large-Scale Time Series Datasets

To confirm that the PS-WFCMdd method is also effective on large-scale datasets in Table 2, in this section, we compare the clustering effects of PS-WFCMdd and six other clustering methods on the eight datasets in Figure 3, Figure 4 and Figure 5 show the ARI scores, FMI scores, NMI scores of the seven clustering methods on each dataset.

By looking at Figure 3. We can see that on the ChlorineConcentration, Rock, and Strawberry datasets, the PS-WFCMdd algorithm has obvious score advantages. On the ECG5000, HouseTwenty, and Wafer datasets, the ARI score of the PS-WFCMdd algorithm is also the best. On the StarLightCurves dataset, the ARI score of the PS-WFCMdd-sD algorithm is the best. On the Mallat dataset, the ARI score of the K-shape algorithm is the best. The ARI scores of the PS-WFCMdd-sD and PS-WFCMdd algorithm on the ChlorineConcentration and HouseTwenty data have little difference.

According to the results in Figure 4. On the Rock, StarLightCurves, and Strawberry datasets, the PS-WFCMdd algorithm has obvious advantages. On the ECG5000 and Mallat datasets, the PS-WFCMdd algorithm is barely optimal. The FCM algorithm achieves excellent performance on the ChlorineConcentration dataset, and the PS-K-means algorithm achieves the highest score on the HouseTwenty dataset. On the Wafer dataset, the FMI score of the PS-WFCMdd-sD algorithm is outstanding.

It can be seen from Figure 5 that on the ChlorineConcentration, Rock, StarLightCurves, Strawberry, and Wafer datasets, the PS-WFCMdd algorithm has obvious score advantages; on the HouseTwenty dataset, the PS-WFCMdd algorithm achieves only a small advantage; on the ECG5000 dataset, the NMI score of the FCM algorithm is the optimal value; on the Mallat dataset, the K-shape algorithm has the best FMI score.

From the above analysis, it can be concluded that our algorithm can achieve excellent performance under three evaluation indicators when faced with large-scale time series datasets. On some datasets, the FCM algorithm and the K-shape algorithm have also achieved good results, indicating that these two algorithms have certain advantages in processing large-scale time series data. The PS-WFCMdd-sD algorithm has achieved good performance under the three evaluation indicators, but the overall performance is still better than the PS-WFCMdd algorithm.

### 4.3. Experimental Analysis from the Perspective of Data Characteristics

To more intuitively explain the robustness of the PS-WFCMdd algorithm from the data features, the trend of all experimental data and the cluster center results obtained by the PS-WFCMdd algorithm are shown in Figure 6. The K-means-sD algorithm, which is generally used in time series clustering, is selected for the corresponding visual comparison, as shown in Figure 7. In both figures, the colored lines represent cluster centers, and the light-colored lines represent the original sample set. In addition, the comprehensive performance of K-means-sD in the experiments in this paper is also relatively good. Let us first observe the clustering centers in the two visualization charts. Through comparison, we can see that since k-means-sd is the clustering center gradually generated in the clustering process, the trend of the clustering centers of some data is quite different from the original data, and the difference is more obvious in BeetleFly, HouseTwenty, and Wafer. However, PS-WFCMdd is the cluster center found in the corresponding original data, and the trend of the obtained cluster center is the same as that of the original data.

We then evaluate the data from the trend in the time series dynamic analysis. From the display of the source data set alone, it can be seen that BeetleFly, Car, Fish, Herring, and ChlorineConcentration all exhibit periodic changes, among which BeetleFly has a larger time series amplitude. In terms of the experimental results on Fish and Herring, PS-WFCMdd performs generally in the three evaluation indicators, while K-means-sD has better experimental results. In the experimental results of Car and ChlorineConcentration, the advantage of PS-WFCMdd is small. PS-WFCMdd performs well in the experimental results on BeetleFly. This shows that when faced with data with obvious periodicity and stable data trend, because there are few noise and abnormal samples in these data, PS-WFCMdd has no advantage in experimental results. The remaining 13 datasets contain a large number of irregular dynamic time series, which inevitably contain a lot of noise and edge data. From the experimental results of these data, the performance of PS-WFCMdd is relatively good. However, on Housetwenty and Wafer data with mutation frequencies, this type of data often presents more difficult modeling problems. PS-WFCMdd performs best among all comparison algorithms, but is not outstanding and does not make a practical solution for mutation frequency time series.

Based on the above experimental analysis, P-splines are used to fit time series data and use the fitted data as the input of clustering, which can effectively improve the final clustering effect. Under the influence of the new weight, the PS-WFCMdd algorithm can still obtain a higher clustering effectiveness index score under the condition of noise and outlier samples in the sample data. However, our algorithm also has some limitations. For example, compared with most algorithms, PS-WFCMdd needs to adjust more smoothing parameters, so the algorithm is not very concise. When faced with time series with obvious periodicity and stable data trends and irregularly changing data with mutation, our algorithm performs in general and needs further optimization and improvement.

### 4.4. Time Complexity Analysis

The time complexity of each comparison algorithm and PS-WFCMdd algorithm in this experiment is shown in Table 6, where *N* represents the number of sample data, *C* represents the number of clusters, and *K* represents the number of iterations. It can be seen that the time complexity of PS-WFCMdd is the same as that of K-shape, and it is second only to FCM and PS-K-means algorithm. The time complexity of the remaining clustering algorithms is higher. We can see that because of the difference in time complexity between soft-DTW and MASS 3, the running time consumption of PS-WFCMdd-sD and PS-WFCMdd is one order of magnitude different. Especially in the face of large time series, the advantages of the PS-WFCMdd algorithm are more obvious.

## 5. Conclusions

This paper proposes a clustering method called PS-WFCMdd suitable for time series characterized by being fast, parsimonious, and noise-resistant. The idea behind our proposal is simple and effective. It can solve the effect of noisy points and outlier samples on fuzzy clustering. We recommend fitting each series with a P-spline smoother before clustering, and using the fitted data as input to the clustering analysis. We also define a new weighting method to avoid the impact of outlier sample points on medoids update. The new weight allocation makes the selected new medoids more appropriate. In the experiment, compared with soft-DTW, MASS 3 has a better advantage in our new weighted fuzzy C-medoids method, MASS 3 can measure the similarity between time series data and medoids more accurately and quickly. In addition, the performance of our weighted fuzzy C-medoids method based on P-spline has been analyzed through three clustering evaluation indexes in Section 4. Our experimental results show that our algorithm can ensure high-quality clustering performance within a reasonable computing time, whether on ordinary time series data sets or large time series data.

In future work, we will try to study a better cluster initialization method to reduce the number of iterations. The distance measurement criterion is further improved to reduce the time complexity of the algorithm, to better improve the comprehensive performance of the algorithm. There are still many open-ended problems that need to be solved. The characteristics of each time series dataset are different. To obtain higher clustering accuracy, it is necessary to analyze the characteristics of the data more carefully and study corresponding characteristics. The corresponding clustering method is obtained, and a more appropriate similarity measure criterion is selected. These considerations can be gradually realized in future research, which must be very interesting. Finally, the code URL for PS-WFCMdd is: https://github.com/houqinchen/PS-WFCMdd (accessed on 1 February 2022).

## Figures and Tables

**Figure 1 sensors-22-06163-f001:**
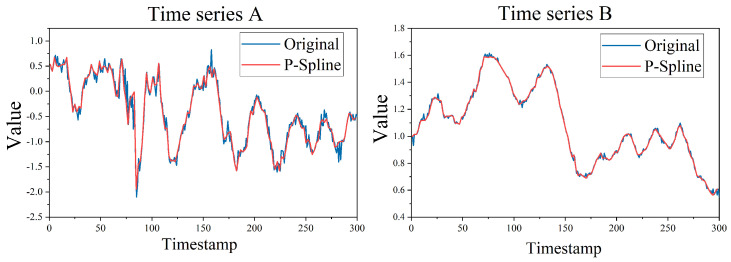
Comparison of P−spline fitting processing results.

**Figure 2 sensors-22-06163-f002:**
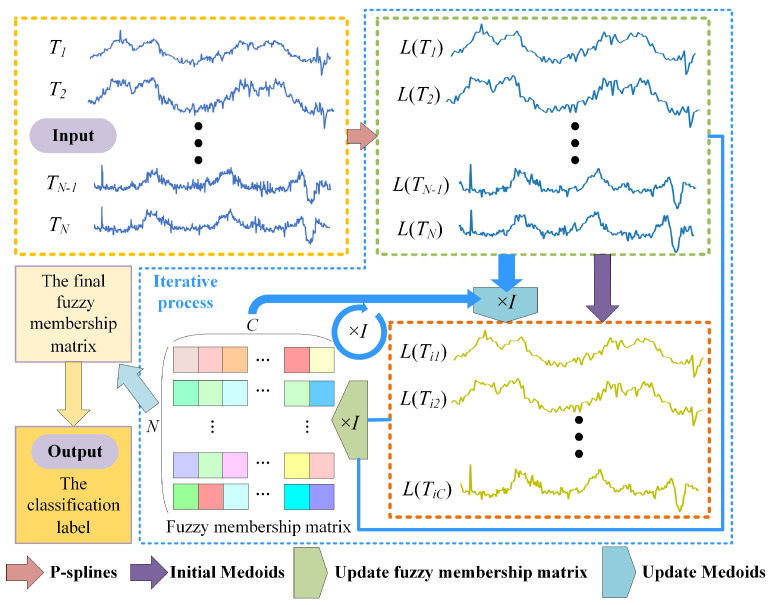
Flow chart of PS−WFCMdd.

**Figure 3 sensors-22-06163-f003:**
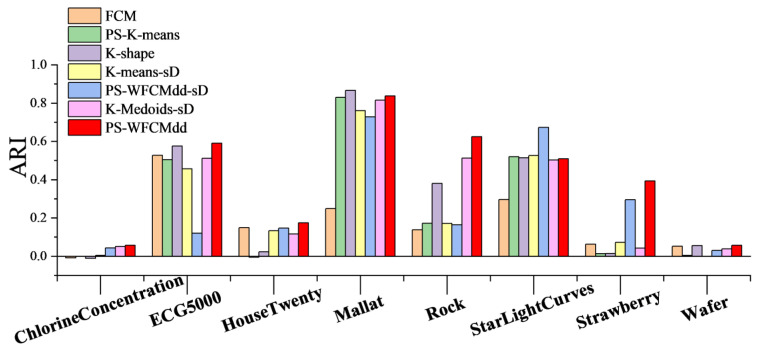
ARI of clustering results with different clustering algorithms.

**Figure 4 sensors-22-06163-f004:**
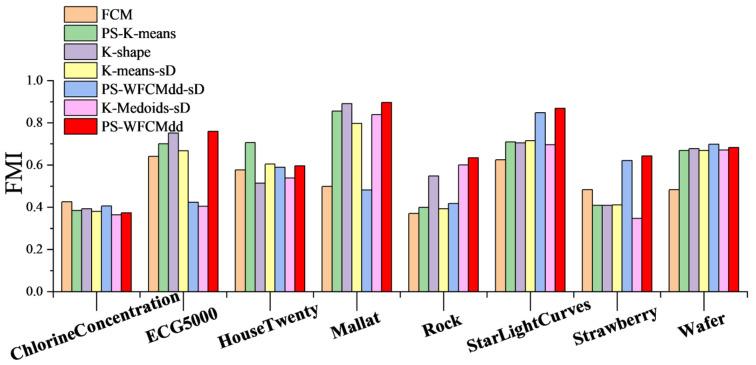
FMI of clustering results with different clustering algorithms.

**Figure 5 sensors-22-06163-f005:**
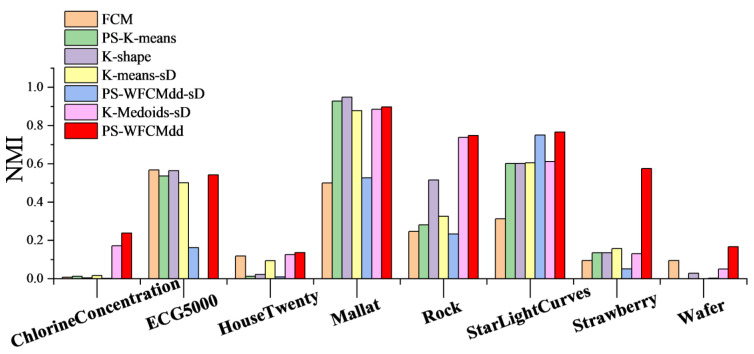
NMI of clustering results with different clustering algorithms.

**Figure 6 sensors-22-06163-f006:**
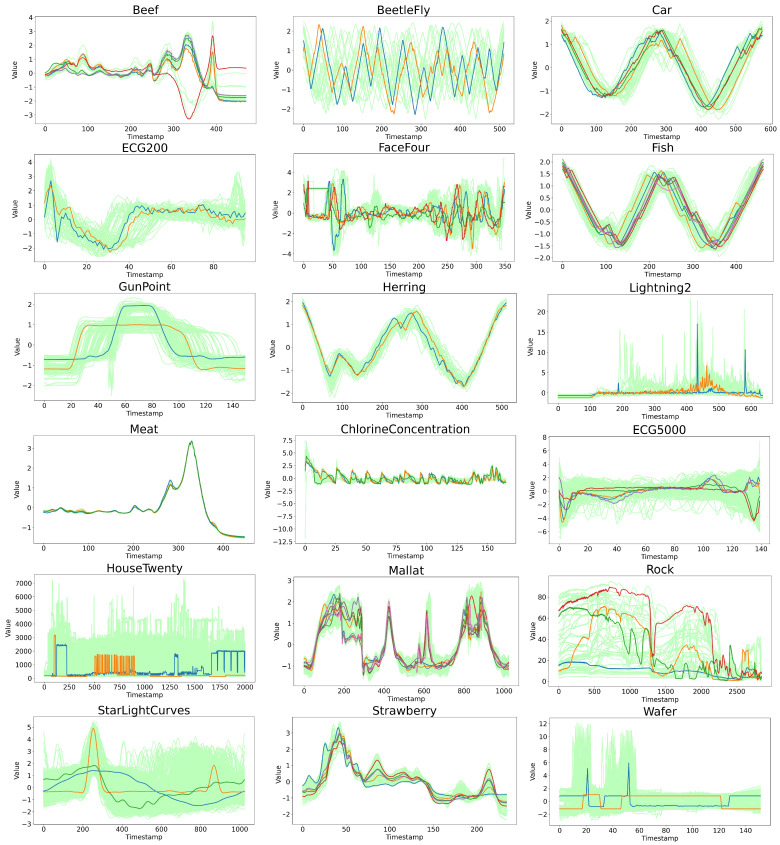
Display the trend of the original data and the cluster centers obtained by the PS−WFCMdd.

**Figure 7 sensors-22-06163-f007:**
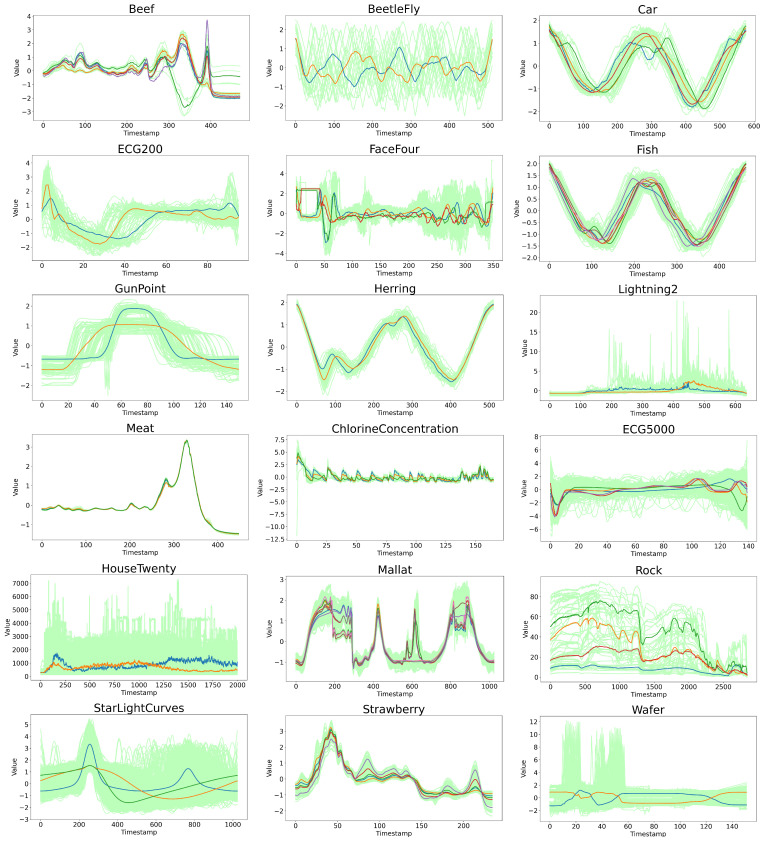
Display the trend of the original data and the cluster centers obtained by the K−means−sD.

**Table 1 sensors-22-06163-t001:** Descriptions of ten common time series data.

Dataset	Number of Classes	Time Series Length	Size of Set
Beef	5	470	60
BeetleFly	2	512	40
Car	4	577	120
ECG200	2	96	200
FaceFour	4	350	88
Fish	7	463	175
GunPoint	2	150	200
Herring	2	512	128
Lightning2	2	637	121
Meat	3	448	120

**Table 2 sensors-22-06163-t002:** Descriptions of eight large-scale time series data.

Dataset	Number of Classes	Time Series Length	Size of Set
ChlorineConcentration	3	166	3840
ECG5000	5	140	4500
HouseTwenty	2	2000	159
Mallat	8	1024	2345
StarLightCurves	3	1024	8236
Strawberry	5	235	983
Rock	4	2844	70
Wafer	2	152	6174

**Table 3 sensors-22-06163-t003:** ARI index scores of different clustering algorithms.

Dataset	FCM	PS-K-Means	K-Shape	K-Means-sD	PS-WFCMdd-sD	K-Medoids-sD	PS-WFCMdd
Beef	0.03229	0.06846	0.01537	0.06846	0.05139	−0.02605	**0.12876**
BeetleFly	0.11333	−0.04442	0.11333	−0.04442	0.03537	0.16197	**0.18711**
Car	0.08753	0.14338	0.09126	0.13345	0.00699	−0.00475	**0.16309**
ECG200	0.00757	0.24208	0.24919	0.28307	0.06660	0.20403	**0.34122**
FaceFour	0.31835	0.26386	0.26957	0.12286	0.36096	0.37089	**0.37165**
Fish	0.21965	0.17013	0.20009	**0.28803**	0.19032	0.17431	0.25790
GunPoint	−0.00512	−0.00512	−0.00512	−0.00512	0.01594	0.00600	**0.15795**
Herring	−0.01435	−0.01488	−0.00852	**0.07876**	0.03936	0.00893	0.04904
Lightning2	0.01194	0.03000	0.00000	0.01194	0.03764	0.03737	**0.03801**
Meat	0.56296	0.49599	0.56296	0.56296	0.75734	0.58745	**0.83168**

**Table 4 sensors-22-06163-t004:** FMI index scores of different clustering algorithms.

Dataset	FCM	PS-K-Means	K-Shape	K-Means-sD	PS-WFCMdd-sD	K-Medoids-sD	PS-WFCMdd
Beef	0.25318	0.26423	0.24121	0.26423	0.40805	0.20412	**0.42276**
BeetleFly	0.53333	0.45305	0.53333	0.45305	0.73536	0.71824	**0.74662**
Car	0.31788	0.37009	0.34238	0.41842	**0.46597**	0.25424	0.43560
ECG200	0.54056	0.68526	0.67556	**0.70917**	0.48946	0.62784	0.66386
FaceFour	0.52085	0.48106	0.47705	0.37907	**0.57572**	0.56173	0.56499
Fish	0.36878	0.30637	0.33106	0.41914	0.37703	0.30304	**0.38720**
GunPoint	0.49524	0.49524	0.49524	0.49524	0.42098	0.49529	**0.51423**
Herring	0.49695	0.50080	0.51599	0.58125	**0.60380**	0.51251	0.54077
Lightning2	0.52741	0.60282	**0.74029**	0.52741	0.52637	0.54414	0.56615
Meat	0.76657	0.66358	0.76657	0.76657	0.75638	0.72042	**0.88439**

**Table 5 sensors-22-06163-t005:** NMI index scores of different clustering algorithms.

Dataset	FCM	PS-K-Means	K-Shape	K-Means-sD	PS-WFCMdd-sD	K-Medoids-sD	PS-WFCMdd
Beef	0.25365	0.31947	0.24797	0.31947	0.25552	0.21094	**0.39062**
BeetleFly	0.11871	0.00733	0.11871	0.00733	0.33175	0.35321	**0.38343**
Car	0.16604	0.26708	0.22220	0.26604	0.11472	0.06676	**0.31387**
ECG200	0.00576	0.14024	0.15085	0.17160	0.14998	0.17360	**0.23618**
FaceFour	0.37440	0.44330	0.34980	0.24702	0.13612	0.50233	**0.54600**
Fish	0.38036	0.34383	0.37170	**0.40286**	0.23739	0.27090	0.35590
GunPoint	0.00111	0.00111	0.00111	0.00111	0.08702	0.04770	**0.20012**
Herring	0.00075	0.00271	0.00128	0.18105	0.13313	0.13073	**0.18397**
Lightning2	0.01523	0.01026	0.00000	0.01523	0.08381	0.07639	**0.08742**
Meat	0.73368	0.59092	0.73368	0.73368	0.71512	0.64352	**0.83955**

**Table 6 sensors-22-06163-t006:** Time complexity comparison.

Algorithm	Time Complexity
FCM	O(N×C×K)
PS-K-means	O(N×C×K)
K-shape	$O\left(N\times \log(N)\times C\times K\right)$
K-means-sD	$O\left(N^{2}\times C\times K\right)$
PS-WFCMdd-sD	$O\left(N^{2}\times C\times K\right)$
K-Medoids-sD	$O\left(N^{2}\times C\times K\right)$
PS-WFCMdd	$O\left(N\times \log(N)\times C\times K\right)$

## Data Availability

The dataset that support the findings of this study are available in https://www.cs.ucr.edu/~eamonn/time_series_data_2018/ (accessed on 1 February 2022).

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
