# Peer review of "A Fast Weighted Fuzzy C-Medoids Clustering for Time Series Data Based on P-Splines"

_sensors, 2022, doi:10.3390/s22166163_

Round 1

Reviewer 1 Report

Review "A Fast Weighted Fuzzy C-Medoids Clustering for Time Series Data based on P-splines" 

The paper describes a procedure for clustering time series data. The procedure is a combination of existing techniques and additionally features an adjusted weighting method for WFCMdd.

 The problem itself (clustering time series) is definitely relevant, as it is quite a common problem in a lot of different domains. It is also an area of ongoing research, since there exists no solution, that solves all kind of clustering problems adequately.

From an originality perspective, the proposed procedure is not groundbreaking, but still can be an interesting addition to the state-of-the-art.

 The paper itself has a clear structure and the authors make it easy to comprehend their content. Formatting and appearance of tables and figures can be considered high quality. The introduction and background section give a good introduction into the topic. The authors do a good job mentioning the required background information. To evaluate the proposed procedure, the authors conduct experiments on several different time series. Overall, the paper seems quite promising.

 Yet, currently, I still would rate the paper as ‘major revision’ as I’d like to see the following points addressed:

From time to time, there are some simple language and spelling errors in the text (e.g. lines 160, 194, below equation 16). These issues should be easy to fix with another proof-read of the whole document.

In the experiments section, more information about your choices would be useful. For example, information about the choice of the evaluation criteria and the selection of datasets would be really helpful. As there are other widely used metrics, I would be interested to know why you specifically chose ARI, FMI, NMI. For the datasets, it would be interesting to know what the detailed selection criteria for inclusion in the experiments were (beyond the fact that you wanted some smaller and some larger scale datasets). In the experiments section, it would be nice to see a discussion to what extent you think your findings are significant. Also in the experiments section, it would be good to see 4.3 the visual inspection described in more detail. Figure 6 could be explained in more detail. I'd be additionally interested what the visual clustering results with one of the other algorithms looks like.

I also was not able to find a link to your source code in the paper. I think it is important to share the code of your experiments in an online repository and include a link in the paper. On the one hand, this increases trust in the experimental results and on the other hand, this greatly helps to promote your new procedure. Conducting reproducible research is extremely important, and everything to do so is freely available nowadays.

Author Response

Thank you very much for your comments and suggestions. We have discussed and revised point by point. For details, see the attachment.

Reviewer 2 Report

In this paper, the authors proposed a clustering method called PS-WFCMdd suitable for time series characterized with fast, parsimonious, and noise-resistant. The idea behind our proposal is simple and effective. It can solve the effect of noisy points and outlier samples on fuzzy clustering. They recommended fitting each series with a P-spline smoother before clustering, and using the fitted data as input to the clustering analysis. The authors also defined a new weighting method to avoid the impact of outlier sample points on medics update. The new weight allocation makes the selected new medoids more appropriate.  It was evaluated and showed significant results.

The paper is well presented, but it needs more modifications, such as

- Elaborate the main contribution in the intro section, you may discuss the limitations of previous ones and how did you overcome them by the new method.  

- add more details about the proposed method, such as

- Y labels are missed in many figures. Fix this issue.

- Limitations should be discussed.

- parameter settings of all compared methods should be discussed. How did you guarantee fair comparisons?

-The paper lacks to deep discussion for different fuzzy systems used for time series applications, you may read and discuss: Pythagorean fuzzy time series model based on Pythagorean fuzzy c-means and improved Markov weighted in the prediction of the new COVID-19 cases; Particle swarm optimization of partitions and fuzzy order for fuzzy time series forecasting of COVID-19; Modified aquila optimizer for forecasting oil production; High-order fuzzy time series forecasting by using membership values along with data and support vector machine; Boosted ANFIS model using augmented marine predator algorithm with mutation operators for wind power forecasting;

Author Response

(The authors gave the same response as above.)

Reviewer 3 Report

The article should be improved by providing an application to a real data set originated from a problem that concerns the science and technology of sensors and is within the aims and scopes of the Journal.

Moreover, the authors should clarify how (and why) they have selected the datasets used in the empirical study. Is each dataset characterized by time series having different dynamics? how do these differences affect the final performance of the proposed procedure?

Author Response

(The authors gave the same response as above.)

Round 2

Reviewer 1 Report

Thank you for the updated version. Looks good, the only minor thing I'd like to ask for is a final proofread before publishing the paper. Language and style in general already look good, but just to make sure you did not overlook something. E.g. in line 209 I spotted 'equationn'.

Author Response

I respect your careful review very much,we have carefully checked the article and made corrections.

Thanks for your comments.

Reviewer 2 Report

The authors addressed all comments. This paper can be accepted. 

Author Response

Thank you very much for your reply.

Reviewer 3 Report

The article can be published as it is

Author Response

Thank you very much for your reply